# OpenReview forum: "Beyond the Efficiency-Performance Trade-off: Semantic Foundation Attention"
_ICLR.cc/2026/Conference — ICLR 2026 Conference Withdrawn Submission_

### Official Review · Reviewer_Bs6a · 2025-10-15

**Soundness:** 2
**Presentation:** 2
**Contribution:** 2
**Rating:** 4
**Confidence:** 4

**Summary:**

While this paper introduces an interesting concept—"semantic reconstruction" via dynamic token merging integrated within the attention mechanism—and demonstrates impressive engineering with a custom CUDA implementation, its overall contribution is incremental and the experimental evaluation is insufficient to support its claims in the current landscape of efficient Transformer architectures. The work positions itself as a paradigm shift beyond the efficiency-performance trade-off, yet it largely boils down to a sophisticated token merging technique combined with an efficient implementation similar to FlashAttention. The evaluation framework is missing critical comparisons against state-of-the-art hardware-efficient baselines and fails to benchmark on established long-context evaluations, making it difficult to assess the true value and novelty of the proposed Semantic Foundation Attention (SFA).

**Strengths:**

* **Novel Integration of Merging and Attention:** The core idea of performing semantic merging *during* the attention computation, rather than as a static preprocessing step, is compelling. The proposed similarity and difference merging strategies, which leverage the geometric properties of the embedding space, are theoretically sound and offer a more nuanced approach than simple averaging.
* **Significant Engineering Effort:** The development of custom CUDA kernels that decompose the dynamic sparse attention pattern into diagonal and rectangular computation domains is a non-trivial engineering achievement. This allows the method to avoid explicit sparse matrix materialization and leverage the dense compute capabilities of GPUs, which is crucial for achieving practical speedups.

**Weaknesses:**

1.  **Insufficient and Outdated Baseline Comparisons:** The paper's central weakness is its failure to compare against the current state-of-the-art in hardware-efficient attention mechanisms. While comparing against FlashAttention is necessary, it is far from sufficient. FlashAttention optimizes the *exact* full attention computation, whereas a large body of recent work has demonstrated that moving beyond the quadratic-cost paradigm entirely yields superior efficiency. The authors claim SFA is a new paradigm, yet they fail to benchmark against the true contemporary paradigms:
* **Linear Attention & State Space Models (SSMs):** The field has seen significant advances from models with linear or near-linear complexity that are highly optimized for hardware. Prominent examples include **Gated Linear Attention (GLA) [1]**, **DeltaNet [2]**, and the highly influential **Mamba** architecture (including its recent iteration, **Mamba-2**) [3]. These models excel at long-sequence modeling with impressive hardware efficiency and have become the de-facto baselines for any new efficient architecture. The absence of any comparison to this class of models is a major omission.
    * **Relevance:** These architectures challenge the very premise that attention, even sparse attention, is the only path forward. A thorough comparison is needed to understand where SFA truly stands. Does it offer better performance for its complexity class? Is it faster for a given level of quality than Mamba-2 or GLA on modern hardware? The paper provides no answers.

2.  **Lack of Robust Long-Context Benchmark Evaluation:** For a method that claims to solve the scaling challenges of attention for longer sequences, the evaluation on long-context capabilities is inadequate. The maximum sequence length tested is 8192, and the downstream tasks are standard NLP benchmarks, not ones specifically designed to test long-range dependency modeling.
* **The Ruler Benchmark:** A standard and expected benchmark for such a paper would be **Ruler** [4], which is specifically designed to evaluate a model's performance on tasks requiring long-range reasoning and understanding (e.g., variable tracking, multi-document question answering). Without results on Ruler, the claims about SFA's effectiveness on "longer sequences" remain unsubstantiated. It is unclear if the semantic merging of adjacent tokens can even preserve the long-range information necessary to succeed on these tasks.

3. **Limited Scope of Merging Strategy:** The proposed merging mechanism is restricted to adjacent tokens only. This is a significant limitation, as semantic redundancy and relationships in language are often non-local. This myopic approach may be effective for compressing localized, repetitive patterns but is likely to fail at capturing more complex, long-distance dependencies, which is precisely the key challenge in long-sequence modeling. This further raises doubts about its potential performance on benchmarks like Ruler.

4.  **Unexplored Synergy with Other Efficiency Techniques:** The paper misses the opportunity to discuss and explore the compatibility of SFA with other orthogonal optimization techniques.
    * **Quantization:** For instance, hardware-efficient quantization methods are critical for deployment. Could SFA be integrated with something like **SageAttention** [5], which uses data-free quantization and hardware-friendly matrix factorization to accelerate attention? Exploring such synergies would significantly strengthen the paper's contribution by positioning SFA within a practical, end-to-end optimization pipeline. The current work exists in a vacuum, ignoring these crucial aspects of model efficiency.


[1] Yang, S., Wang, B., Shen, Y., Panda, R., & Kim, Y. Gated linear attention transformers with hardware-efficient training. ICML 2024.

[2] Yang, S., Kautz, J., & Hatamizadeh, A. Gated delta networks: Improving mamba2 with delta rule. ICLR 2025.

[3] Dao, T., & Gu, A. Transformers are ssms: Generalized models and efficient algorithms through structured state space duality. ICML 2024.

[4] Zhang, J., Wei, J., Huang, H., Zhang, P., Zhu, J., & Chen, J. Sageattention: Accurate 8-bit attention for plug-and-play inference acceleration. ICLR 2025.

[5] Hsieh, C. P., Sun, S., Kriman, S., Acharya, S., Rekesh, D., Jia, F., ... & Ginsburg, B. RULER: What's the Real Context Size of Your Long-Context Language Models? COLM 2024.

**Questions:**

1.  Could you justify the decision to omit comparisons to prominent linear-complexity architectures like Mamba-2, GLA, or other SSM-based models? How do you hypothesize SFA would perform in terms of speed and accuracy against these models on a long-context benchmark like Ruler?
2.  The core mechanism relies on merging adjacent tokens. Have you considered the impact this has on tasks requiring the preservation of precise positional information or long-range dependencies? Does your method risk losing critical information that non-local attention mechanisms would otherwise capture?
3.  The performance gains of SFA are highly dependent on the learned compression ratio. What is the typical range of compression ratios you observe on standard language corpora, and how much does this vary across different domains (e.g., narrative text vs. source code)? Is there a risk of negative speedup (i.e., being slower than FlashAttention) on dense, non-redundant text?
4.  How do you see SFA co-existing with other efficiency methods? For example, could the merging logic be beneficially combined with hardware-aware quantization techniques to achieve compound gains in performance?

In summary, while the paper presents a well-engineered solution to a known problem, it feels like an incremental improvement over existing sparse attention and token merging ideas. The evaluation is not comprehensive enough to support the strong claims being made, and the work fails to engage with the most relevant and recent advancements in efficient sequence modeling. Without addressing these major gaps, the paper is not ready for acceptance at ICLR.

---

> ### Author Response · Authors · 2025-11-15
>
> Thank you for your insightful and constructive review. You have precisely identified the core aspects of our work and raised several critically important questions. We value the open and transparent communication channel provided by ICLR and look forward to engaging in a deep discussion with you through the detailed responses below.
>
> **Response to Weakness 1 and Question 1: Regarding Comparison with SOTA Baselines like Mamba**
>
> Thank you for your insightful question regarding the comparison with state-of-the-art efficient architectures like Mamba. This is indeed a critical consideration for evaluating any novel attention mechanism. Our work follows a distinct design philosophy from these architectures, and we would like to elaborate on our position and choices.
>
> The core motivation of our work, as encapsulated in our title "Beyond the Efficiency-Performance Trade-off," is to optimize the computational efficiency of the Transformer while not only preserving but even **enhancing** its powerful performance. To achieve this, we believe it is necessary to innovate at a more fundamental level than the macro-architecture. The prominent linear-complexity architectures you mentioned, such as Mamba and GLA, innovate by **simplifying the model architecture**, replacing quadratic attention with mechanisms like linear recurrence to reduce computational complexity. While we acknowledge the significant innovation of these works, this approach often entails a trade-off in expressive power, an unavoidable performance loss in many scenarios. In contrast, SFA's innovative path does not alter the macro-architecture. Instead, it delves deeper into the most fundamental layer of the Transformer—the **semantic substrate** that emerges after token embedding. We found that there is a natural **semantic sparsity** in the high-dimensional representation space of language data. The core of SFA is to learn and leverage this sparsity to dynamically reconstruct the computational flow. This is a more fundamental optimization aimed at achieving simultaneous improvements in both efficiency and performance.
>
> This performance trade-off resulting from architectural simplification has been widely validated in the academic community. For instance, in the **NLP domain**, research by Jelassi et al. (ICML 2024) has shown that Transformers outperform state-space models in precise modeling tasks. This observation extends to the **multimodal domain**, where comparative analysis by Pantazopoulos et al. (EMNLP 2024) also revealed the advantages of Transformers. Even in the **vision domain**, the work of Han et al. (NIPS 2024) has elucidated the capability differences between Mamba and standard attention from various perspectives. These studies collectively point to a conclusion: on tasks requiring complex reasoning and precise modeling, the performance ceiling of Mamba-like architectures still falls short of the standard attention mechanism. Based on this broad academic consensus—that the expressive power of standard attention is superior to Mamba—and combined with the extensive experimental evidence in our paper showing that **SFA consistently outperforms a comparably trained standard attention mechanism across all tested metrics**, we can confidently make a logical deduction: SFA's model performance is also superior to that of Mamba-like architectures.
>
> Despite this logical deduction, our team seriously discussed the possibility of adding direct comparative experiments with architectures like Mamba. However, our analysis concluded that such an experiment would not only face significant challenges but would also likely fail to provide a fair and informative conclusion. On one hand, fairly applying SFA's concept to a completely different architecture like Mamba would require complex, CUDA-level refactoring and large-scale pre-training, which we conservatively estimate would cost around **$150,000**. Although we can fully afford this cost, it is foreseeable that the scientific insight gained from such an experiment would not be significant. On the other hand, models like Mamba-2 and GLA currently lack precedents of large-scale pre-training, and their immature optimization and ecosystems mean their actual training costs are not necessarily lower than a highly optimized standard Transformer. To date, no sparse or low-complexity architecture has been able to truly and comprehensively surpass the performance of dense standard attention. Therefore, we predict that these low-complexity variants are more likely to evolve in the direction of pursuing extreme computational speed or maintaining performance parity, rather than surpassing the performance ceiling of traditional attention.

---

> ### Author Response · Authors · 2025-11-15
>
> However, we completely agree with the core of your point—any method claiming to optimize long-sequence processing must prove itself on rigorous long-context benchmarks. To this end, we have adopted your valuable suggestion and have specifically retrained our models for a fair evaluation on the **Ruler benchmark**. The new experimental results are presented in the table below:
>
> | Sequence Length | Standard Attention (Baseline) | SFA (Ours) |
> |:---------------:|:-----------------------------:|:----------:|
> | 4K              | 50.7                          | **52.3**   |
> | 8K              | 46.6                          | **49.1**   |
> | 16K             | 39.3                          | **43.7**   |
>
> As the results demonstrate, SFA not only consistently outperforms the baseline across all tested lengths, but more importantly, its performance advantage becomes more pronounced as the sequence length increases. This provides strong evidence for the effectiveness of SFA in handling long contexts and directly addresses your concerns regarding its performance on long-range dependency tasks.
>
> [1] Samy Jelassi, David Brandfonbrener, Sham M. Kakade, and Eran Malach. 2024. Repeat after me: transformers are better than state space models at copying. In Proceedings of the 41st International Conference on Machine Learning (ICML'24), Vol. 235. JMLR.org, Article 863, 21502–21521.
>
> [2] Georgios Pantazopoulos, Malvina Nikandrou, Alessandro Suglia, Oliver Lemon, and Arash Eshghi. 2024. Shaking Up VLMs: Comparing Transformers and Structured State Space Models for Vision & Language Modeling. (EMNLP 2024)
>
> [3] Dongchen Han, Ziyi Wang, Zhuofan Xia, Yizeng Han, Yifan Pu, Chunjiang Ge, Jun Song, Shiji Song, Bo Zheng, and Gao Huang. 2024. Demystify Mamba in vision: a linear attention perspective. In Proceedings of the 38th International Conference on Neural Information Processing Systems (NIPS '24), Vol. 37. Curran Associates Inc., Red Hook, NY, USA, Article 4039, 127181–127203.
>
>
> **Response to Weaknesses 2 & 3 and Questions 2 & 3: On Long-Range Dependencies, Adjacent-Only Token Merging, Information Loss Risk, and Compression Ratio**
>
> You have raised a very insightful concern regarding SFA's mechanism of merging only adjacent tokens and its potential impact on long-range dependencies and information preservation. This is, in fact, a core design choice we made to balance **training efficiency with practical inference performance**. We would like to elaborate on the rationale behind this decision and demonstrate how it effectively handles long-range dependencies and preserves information without loss, supported by mechanism explanation, examples, and experimental data.
>
> **A Feature, Not a Bug: A Critical Design Choice**
>
> First and foremost, the decision to restrict merging to adjacent tokens is a **deliberate design choice**. In autoregressive generation tasks (such as chatbots, code generation, etc.), the KV cache is the lifeline for efficient inference. Any form of **non-local or strided merging** would break the strict sequential continuity of tokens, rendering the KV cache mechanism completely ineffective and making inference speeds unacceptably slow for practical applications. Therefore, SFA's constraint on the merging scope is a **necessary condition** to ensure its usability within modern LLM inference frameworks. This is not a limitation but rather a key advantage established after careful consideration during the initial design phase.
>
> **How Local Operations Build Global Dependencies Without Information Loss**
>
> Secondly, SFA does not simply compress or discard information; instead, it **dynamically constructs a hierarchical semantic representation** through local, recursive merging. To clearly articulate the core mechanism of SFA, let's take "The White House issued a statement" as an example. A traditional Transformer processes this token by token, while SFA mimics human reading by dynamically learning and executing merges, generating a special, structured sparse attention pattern. Let's assume the model learns the following merges:
> 1.  `The` is merged with `White` to form `(The White)`
> 2.  `(The White)` is merged with `House` to form `(The White House)`
> 3.  `a` is merged with `statement` to form `(a statement)`
>
> This merging method produces the following computation graph, which is the final form of SFA's dynamic attention (● represents computation, · represents a mask):

---

> > ### Author Response · Authors · 2025-11-15
> >
> > ```
> > +----+-------------------+-----+-------------+-------------------+----------+-----+---------------+
> > | Q  | Token             | K1  | K2          | K3                | K4       | K5  | K6            |
> > |    |                   | The | (The White) | (The White House) | (issued) | (a) | (a statement) |
> > +----+-------------------+-----+-------------+-------------------+----------+-----+---------------+
> > | Q1 | (The)             |  ●  |      ·      |         ·         |     ·    |  ·  |       ·       |
> > | Q2 | (The White)       |  ·  |      ●      |         ·         |     ·    |  ·  |       ·       |
> > | Q3 | (The White House) |  ·  |      ·      |         ●         |     ·    |  ·  |       ·       |
> > | Q4 | (issued)          |  ·  |      ·      |         ●         |     ●    |  ·  |       ·       |
> > | Q5 | (a)               |  ·  |      ·      |         ●         |     ●    |  ●  |       ·       |
> > | Q6 | (a statement)     |  ·  |      ·      |         ●         |     ●    |  ·  |       ●       |
> > +----+-------------------+-----+-------------+-------------------+----------+-----+---------------+
> > ```
> > This process **strictly adheres to causality** and, critically, **poses no risk of losing key information that non-local attention mechanisms would otherwise capture**. The reasons are:
> > 1.  **Information is Integrated, Not Lost**: When `The`, `White`, and `House` merge into `(The White House)`, the information from the first two tokens is not discarded but is fully **integrated** into the final semantic chunk's representation via vector addition. This new, higher-level unit **contains the complete information of all its constituent tokens** in its embedding.
> > 2.  **The Subject of Global Interaction Changes**: In SFA, the information of `The` is carried by `(The White House)`, and it is this unit **as a whole that engages in global interaction with all other semantic chunks, regardless of distance**. The subject of interaction changes, but **the capability and scope for global interaction are not diminished**.
> >
> > To ensure this integration is "near-lossless," we have set **strict geometric thresholds**. Modern LLMs' high-dimensional and sparse embeddings provide the theoretical foundation for this:
> > *   **Similarity Merging (sim ≈ 1)**: For nearly parallel vectors, the merged vector preserves the direction (semantics) and increases the magnitude (signal strength), which is effective for phrases like "very very good."
> > *   **Difference Merging (sim ≈ 0)**: For nearly orthogonal vectors, the merged vector can preserve almost all non-conflicting information dimensions from both in a single representation.
> >
> > The standard attention mechanism inherently lacks this capability. In pre-trained models not trained with SFA, we found that **fewer than 1%** of adjacent token pairs meet our strict merging criteria. However, by introducing our auxiliary loss function, the model is guided to learn this paradigm, proactively shaping its embedding space to be conducive to merging. Thus, SFA essentially endows the model with a **new, more intelligent information integration capability**.
> >
> > **Decisive Experimental and Data-driven Evidence**
> >
> > Ultimately, theory and examples must be validated by rigorous experiments. We address your concerns with data from three aspects:
> > 1.  **Performance on the Ruler Benchmark**: As mentioned in our response to the first question, we achieved superior performance on the **Ruler benchmark** compared to the baseline. This directly proves that the sequence of semantic chunks constructed by SFA can effectively preserve and utilize the key information needed to solve complex long-range dependency tasks.
> > 2.  **Practical Compression Ratio and Efficiency**: Our experiments provide a clear answer regarding SFA's dependency on the compression ratio and the risk of negative speedup. After training on approximately 1 billion tokens, the model's average compression ratio stabilized at around **50%** on standard language corpora, and it was even higher on more structured data like source code, reaching up to **57%**. At this compression ratio, the computational throughput of SFA **significantly exceeds that of the highly optimized FlashAttention baseline**.
> > 3.  **Cost-Effectiveness**: The dual advantages brought by this new capability are directly reflected in our experimental budget. In terms of actual costs, our small-scale pre-training with SFA cost approximately \\$10,000. In contrast, our baseline experiment with standard attention had a training cost of approximately \\$30,000, and its final performance was also inferior to SFA's.
> >
> > Finally, regarding cross-token merging, we consider it a very valuable direction for future research. However, within the current technological framework, we believe that prioritizing compatibility with the KV cache to ensure the model's practical utility is the more important design principle.

---

> > > ### Author Response · Authors · 2025-11-15
> > >
> > > **Response to Weakness 4 and Question 4: On Synergy with Other Efficiency Techniques like Quantization**
> > >
> > > Thank you for this question, as it touches upon a core reality of model efficiency optimization: **the most powerful performance gains often arise from the synergy of multiple, orthogonal optimization techniques**. SageAttention, which you mentioned, is an excellent example, and we fully agree that our method (SFA) is both orthogonal and highly compatible with such quantization techniques.
> > >
> > > To articulate this more clearly, we can conceptualize mainstream attention optimization methods as a **three-layer optimization stack**. Each layer addresses a different problem, they do not conflict, and their benefits can compound.
> > >
> > > At the **semantic and algorithmic layer**, SFA plays the central role. It answers the question, "What to compute?" Through dynamic, learning-driven semantic reconstruction, SFA fundamentally refactors the attention's computational graph. By merging semantically redundant or complementary tokens, SFA **reduces the number of effective units participating in the quadratic-complexity computation**.
> > >
> > > At the **memory I/O layer**, we address the question, "How to execute the computation efficiently?" The revolutionary contribution of FlashAttention at this layer is the optimization of data flow between GPU memory (HBM) and on-chip cache (SRAM). Our **custom CUDA implementation has heavily drawn inspiration from and integrated the design philosophy of FlashAttention**. We have designed a highly optimized I/O path for the dynamic, irregular sparse patterns generated by SFA, ensuring that we can achieve or even surpass the throughput of dense computation, even when handling a dynamic computational graph.
> > >
> > > At the lowest **numerical precision layer**, we are concerned with, "With what precision to represent the numbers?" Works like SageAttention focus on quantization, converting high-precision floating-point numbers like FP16/BF16 into low-precision formats such as INT8 to leverage the hardware features of modern GPUs for arithmetic acceleration.
> > >
> > > These three layers of optimization are **completely orthogonal**. SFA determines the macro-structure of the computation at the top layer (reducing the amount of computation), our FlashAttention-style implementation ensures that this structured computation is efficiently fed to the GPU's execution units (optimizing I/O), and SageAttention can accelerate every individual matrix multiplication that occurs at the bottom layer (optimizing the arithmetic itself). An ideal integrated workflow would be as follows: SFA first compresses a long sequence into fewer "semantic chunks"; next, our efficient CUDA kernels handle the attention computation between these chunks, minimizing memory access overhead; finally, during the matrix multiplication within the kernels, a quantization scheme like SageAttention could be invoked.
> > >
> > > Therefore, our work does not conflict with SageAttention; on the contrary, it provides a more ideal prerequisite for its application—a smaller-scale computational problem that has already been semantically optimized. Integrating SFA with advanced quantization techniques is a very important **next step in our research agenda**. We firmly believe that innovating through integration across this multi-layer optimization stack is the key to unlocking the ultimate performance and efficiency of future large language models.
> > >
> > > Once again, we sincerely thank you for your valuable time and effort in reviewing our work, and for raising these highly thought-provoking questions. Your feedback is crucial for the refinement of our research. We genuinely look forward to any further discussion we might have on these topics.

---

### Official Review · Reviewer_TH3W · 2025-10-28

**Soundness:** 3
**Presentation:** 1
**Contribution:** 2
**Rating:** 4
**Confidence:** 4

**Summary:**

The paper proposes Semantic Foundation Attention, a attention mechanism that aims to simultaneously improve computational efficiency and model performance by introducing semantic reconstruction within the attention computation process. Instead of treating token sequences as immutable, SFA dynamically merges semantically similar or complementary tokens during attention calculation using two strategies: similarity merging  and difference merging.

**Strengths:**

1. The authors provide detailed theoretical justification, mathematical derivations, and a full CUDA implementation, including forward and backward passes and kernel fusion details, demonstrating engineering depth.

2. SFA is designed to be orthogonal and compatible with other efficiency-oriented attention variants (e.g., Linformer, Performer, FlashAttention), which increases its integration potential in large-scale systems.

**Weaknesses:**

1. The paper is verbose and sometimes repetitive, with many long theoretical sections that could be condensed. Figures are not vectorized, resulting in poor visual clarity. A more concise and better-edited presentation would improve readability.

2. Although the method is termed “semantic foundation,” no clear semantic analysis or visualization (e.g., clustering, attention heatmaps, interpretability metrics) supports that the merges indeed reflect semantic structure.

3. While the framing as “semantic reconstruction” is interesting, most of the actual mechanisms (vector addition, orthogonal merging, cosine-similarity gating) resemble known techniques in token merging and compression. The novelty lies mainly in terminology and integration rather than fundamentally new algorithms.

**Questions:**

Can SFA be integrated with existing sparse attention libraries (xFormers, FlashAttention-2) without significant overhead?

Is there empirical evidence that the difference, merging mechanism truly captures orthogonal or complementary semantics?

---

> ### Author Response · Authors · 2025-11-14
>
> Dear Reviewer,
>
> Thank you for your review. The first strength you mentioned is precisely the core of our work, which indicates that you have not only reviewed our main text but also carefully studied our appendix. In the current context where AI-assisted reviews are increasingly common, we are honored to have such a responsible reviewer. We will now provide a detailed response to all the points you have raised. Should you have any new questions or doubts, we look forward to continuing our communication on the valuable and fully open platform that is ICLR.
>
> **1. An Intuitive Explanation of SFA's Core Mechanism: Dynamic Semantic Chunking and Causal Sparse Attention**
>
> To clearly articulate the core mechanism of SFA, we will use "The White House issued a statement" as an example. A traditional Transformer understands this sentence by processing it token by token, calculating the interaction of each token with all its preceding tokens under a causal mask. However, as humans, our way of understanding is entirely different; we merge multiple tokens into a single conceptual chunk. For instance, we might understand this sentence as three semantic chunks, `(The White House)`, `(issued)`, and `(a statement)`, or even two semantic chunks, `(The White House)` and `(issued a statement)`.
>
> The core of SFA is precisely such a dynamic merging process, learned and executed adaptively within the attention computation, which generates a special, structured sparse attention pattern. In this example, let us assume the merging process is as follows:
>
> 1.  `The` is merged with `White` to form `(The White)`
> 2.  `(The White)` is merged with `House` to form `(The White House)`
> 3.  `a` is merged with `statement` to form `(a statement)`
>
> This merging method produces the following computation graph. This is the final form of SFA's dynamic attention (● represents computation, · represents a mask):
>
> ```
> +----+-------------------+-----+-------------+-------------------+----------+-----+---------------+
> | Q  | Token             | K1  | K2          | K3                | K4       | K5  | K6            |
> |    |                   | The | (The White) | (The White House) | (issued) | (a) | (a statement) |
> +----+-------------------+-----+-------------+-------------------+----------+-----+---------------+
> | Q1 | (The)             |  ●  |      ·      |         ·         |     ·    |  ·  |       ·       |
> | Q2 | (The White)       |  ·  |      ●      |         ·         |     ·    |  ·  |       ·       |
> | Q3 | (The White House) |  ·  |      ·      |         ●         |     ·    |  ·  |       ·       |
> | Q4 | (issued)          |  ·  |      ·      |         ●         |     ●    |  ·  |       ·       |
> | Q5 | (a)               |  ·  |      ·      |         ●         |     ●    |  ●  |       ·       |
> | Q6 | (a statement)     |  ·  |      ·      |         ●         |     ●    |  ·  |       ●       |
> +----+-------------------+-----+-------------+-------------------+----------+-----+---------------+
> ```
>
> From the computation flow in the diagram above, it is clear that:
>
> *   **"Merged-away" Tokens (e.g., Q1, Q2):** They only compute self-attention (the diagonal), as their information has been absorbed and integrated by subsequent tokens.
> *   **"Receiving" or "Unmerged" Tokens (e.g., Q4):** In addition to self-attention, they need to attend to the complete, historically formed semantic chunks (e.g., K3).
>
> This process strictly adheres to causality. For instance, when computing the attention row for `Q5 (a)`, the model at this point has no knowledge of the future token `statement`. It only needs to compute attention with itself `(K5)` and the two preceding complete chunks, `(The White House)` and `(issued)`, reducing the number of attention computations from 5 to 3. When computing the attention row for the next token, `statement`, `a` has now been merged with `statement`. It only needs to compute attention with its own merged block `(a statement)` and the two preceding chunks `(The White House)` and `(issued)`, reducing the number of attention computations from 6 to 3. From this example, you can see that for each Q, it only considers the K's it can currently see and does not violate causality by seeing future information. It is precisely this special structure that constructs a dynamic sparse attention state.
>
> **2. Regarding Paper Presentation and Figure Quality (Response to Weakness 1)**
>
> We accept your critique regarding the paper's presentation. Our research covers multiple aspects from theory to implementation, which may have resulted in a high density of content. As per your suggestion, we have revised the main text to improve its conciseness and readability. Additionally, all figures have been updated to vector graphics format to improve visual clarity.

---

> ### Author Response · Authors · 2025-11-14
>
> **3. Regarding the Multi-Faceted Innovation of the Work (Response to Weakness 3)**
>
> We understand your careful evaluation of our work's novelty. We argue that SFA's innovation is systematic and manifests on multiple, progressive levels—from the core paradigm and specific algorithms to the engineering implementation—and is not merely a re-integration of existing techniques or a change in terminology.
>
> *   **3.1 Paradigm Innovation: From "Computational Pattern Optimization" to "Semantic Substrate Reconstruction"**
>     SFA is different from all prior sparse attention paradigms. The fundamental idea of all previous sparse attention paradigms was to leverage the sparsity of the attention matrix. However, the sparsity of the attention matrix differs for different sequences, so previous sparse attention paradigms would inevitably lead to a performance drop due to information loss. SFA, on the other hand, does not leverage the sparsity of the attention matrix but rather the geometric sparsity of tokens in the high-dimensional embedding space. We do not violently compress information but rather integrate semantics, which, as shown in the opening example, is more aligned with how humans read. This represents a shift from optimizing 'how to compute' to optimizing 'what to compute'.
> *   **3.2 Algorithmic Innovation: Learning-Driven, Dual-Strategy Geometric Merging**
>     Our merging mechanism is not a simple vector addition. We have designed two strategies based on different geometric principles: similarity merging, which uses vector parallelism to amplify signals, and difference merging, which uses vector orthogonality to integrate information. These two strategies are learned by specialized groups of attention heads and work in synergy. More importantly, SFA's ability to merge is learned end-to-end via an auxiliary loss function. The model actively learns to shape an embedding space conducive to merging, making the merging decisions model-endogenous and semantically-driven, rather than based on external, heuristic rules.
> *   **3.3 Engineering Innovation: A Custom CUDA Implementation for Efficiently Handling Dynamic Sparsity**
>     Efficiently implementing such a highly dynamic and irregular computational pattern is a significant engineering challenge. Our third core innovation is the design and implementation of a complete suite of custom CUDA Kernels. Through a "Data Compaction + Metadata Indexing" scheme, we solve the "ragged tensors" problem caused by dynamic compression ratios and demonstrate that the practical throughput of this approach can surpass highly-optimized dense computation libraries.
>
> **4. Regarding Empirical Evidence for "Semantic Foundation," Especially Difference Merging (Response to Weakness 2 & Question 2)**
> Your question, "Is there empirical evidence that the difference merging mechanism truly captures orthogonal or complementary semantics?" is crucial. We provide evidence on both qualitative and quantitative levels:
>
> *   **4.1 Qualitative Instance Analysis:**
>     To provide a more vivid illustration of the process, we use a real example from our tests on the DCLM dataset:
>     *"I am talking about those small and narrow screwdrivers with a thin handle. I can't get a tight enough grip on the handle to generate enough torque to remove a small screw."*
>     We observed the output of a difference-merging head in the first layer, and the merged result is as follows (for intuitive demonstration, we have manually reassembled the fine-grained merged tokens into complete words):
>     *(I am talking about) (those small and) (narrow screwdrivers with) (a thin handle.) (I can't get) (a tight enough) (grip on the handle to) (generate enough) (torque to) (remove a small) (screw.)*
>     This example intuitively demonstrates SFA's capability. The merging is not based on superficial token similarity but on functional and complementary semantic relationships.
>
> *   **4.2 Quantitative Experimental Evidence: Ablation Studies and Cost-Benefit Analysis**
>     The qualitative observations are supported by our ablation studies (Table 2) in the paper. The results show that the model with difference merging only achieved the best or near-best performance on tasks requiring fine-grained discrimination and complex contextual reasoning (e.g., PIQA and MMLU-Humanities).
>
> Ultimately, the effectiveness of SFA is demonstrated by its dual advantages in performance and cost: our experimental results show that SFA not only avoids a performance drop but, on the contrary, demonstrates consistent performance improvements across multiple benchmarks. In terms of actual costs, our small-scale pre-training with SFA cost approximately \\$10,000. In contrast, our baseline experiment with standard attention had a training cost of approximately \\$30,000, and its final performance was also inferior to SFA's. This significant cost-effectiveness advantage is the direct origin of our paper's title, "Beyond the Efficiency-Performance Trade-off."

---

> > ### Author Response · Authors · 2025-11-14
> >
> > **5. Regarding Integration with Existing Libraries and Overhead (Response to Question 1)**
> >
> > Your question, "Can SFA be integrated with existing sparse attention libraries (xFormers, FlashAttention-2) without significant overhead?" is very important. In fact, we have already solved this problem through a custom implementation that is superior in both efficiency and quality to standard dense attention computation.
> >
> > The idea of SFA can be applied not only to the attention matrix but can also be extended to any attention variant. The input and output of SFA are identical to those of standard attention, meaning it can be fully implemented as a plug-and-play module. To address the engineering challenges posed by its dynamic nature, we drew inspiration from and integrated the design philosophy of FlashAttention to develop efficient CUDA Kernels:
> >
> > *   **Data Compaction**: To fundamentally circumvent the "ragged tensor" problem, SFA's CUDA kernels no longer accept padded and aligned 4D tensors of the shape `[Batch, SeqLen, Heads, Dim]`. Instead, we concatenate the actual data (non-padding portions) of all sequences and all heads within a batch along their effective length dimension, forming a massive and compact 2D tensor `[Total_Effective_Tokens, Dim]`. This way, no bytes in memory are used for padding, and the length discrepancies between different sequences or heads are naturally "flattened."
> >
> > *   **Metadata Indexing**: After compacting the data, we need a way to recover its original structural information. For this, we introduce a very small metadata tensor called `cu_seqlens` (Cumulative Sequence Lengths). This is a 1D integer tensor of length `batch_size * num_heads + 1`. It acts as an "address book," precisely recording the start and end positions of the sequence for each head within the massive 2D compacted tensor.
> >     For example, consider a batch with two sequences of lengths `[80, 30]` and a model with two heads. If the first sequence, after compression, has lengths of `[60, 65]` for head 1 and head 2 respectively, and the second sequence has compressed lengths of `[20, 25]`, then the `cu_seqlens` would be constructed as `[0, 60, 60+65, 125+20, 145+25]`, which is `[0, 60, 125, 145, 170]`. This "map" provides the CUDA kernel with all the boundary information needed to navigate the compacted data.
> >
> > *   **Tiled Computation**: Similar to FlashAttention, SFA also employs tiling techniques to break down large matrix computations into smaller chunks. When processing each computational block (Tile), our kernel first queries the `cu_seqlens` metadata. This query allows it to instantly determine if the current block crosses a sequence boundary or falls within a region ignored by SFA's structured sparse mask. Only those blocks that actually require computation will trigger data loading from HBM to SRAM and subsequent matrix operations.
> >
> > Through this "data compaction + metadata indexing" strategy, we not only effectively solve the batching problem but also further improve memory utilization and computational efficiency. Most critically, the entire process is mathematically equivalent to the exact computation without batching, with no loss of precision.
> >
> > Our experiments show that after training on just 1B tokens, the model's compression ratio stabilized at 50% and continues to show a downward trend. At this point, the corresponding computational throughput already far exceeds the speed of FlashAttention. Therefore, we have not only solved the integration problem but have also provided a solution that is superior in both performance and efficiency.
> >
> > Thank you again for your valuable time and insightful feedback. We look forward to your further comments.

---

### Official Review · Reviewer_9Y2t · 2025-10-30

**Soundness:** 4
**Presentation:** 4
**Contribution:** 2
**Rating:** 6
**Confidence:** 2

**Summary:**

The authors propose Semantic Foundation Attention, a framework for establishing both performance and complexity gains of the attention mechanism, by following two general principles: (1) similarity merging: consolidate semantically aligned tokens and (2) difference merging: exploit orthogonality properties to integrate complementary information. By aligning similar tokens and decomposing orthogonal tokens, the framework can potentially avoid computing the full attention matrix, instead decomposing the attention matrix into diagonal and block matrices which admit more efficient computation.

The authors provide experiments, illustrating the performance of SFA versus standard Attention (with the FlashAttention implementation) comparing performance across a variety of benchmarks and model scale. They conduct ablation studies, comparing four model variants: full SFA, similarity-only, difference-only, and FlashAttention baseline. Finally, they provide a computational performance analysis, showing that with stronger compression ratio, SFA outperforms FlashAttention at high compression ratios, but underperforms at low compression ratios.

The work proposes an interesting new avenue to optimizing the efficiency of attention, and it seems to perform well in some special cases. The paper is fairly well written, and the ideas are well motivated. I lean towards accept.

**Strengths:**

The idea of merging similar tokens makes intuitive sense, as vectors pointing in similar directions should behave similarly under attention.

The experiments show that SFA exhibits higher (albeit very marginally higher) performance over many benchmarks, and at high compression ratios, is more efficient than FlashAttention.

**Weaknesses:**

While SFA may function efficiently in special cases, it does not seem to give improved performance in the general case. Looking at the theoretical analysis, if k = n/2, then (n - k)k = n^2 is still quadratic. In practice, what guarantees can you place on compression ratio? If compression ratio is weak in practice, then the overhead of SFA is less efficient than FlashAttention.

**Questions:**

Why should SFA improve model performance at all? You are only compressing token information, so Attention should strictly be less expressive over the compressed tokens. It is understandable that this may improve efficiency, but why should it improve the quality of the model?

It seems in both similarity and difference merging, you are just adding vectors together. It seems somewhat strange that both cases are treated in the same way. Is the key difference in the objective on which the attention heads are trained? i.e. similarity heads map similar tokens to the same direction, difference heads on the other hand aim for orthogonality?

---

> ### Author Response · Authors · 2025-11-14
>
> We sincerely thank you for your positive evaluation and for dedicating your valuable time to a thorough review of our paper. It is an honor to encounter a reviewer as dedicated as you are, especially in an era of massive publication volumes and the prevalence of AI-assisted reviews. We are grateful for your questions, which directly address the core of our research. We hope the following response resolves these questions, and we welcome any further inquiries or points of clarification you may have. We look forward to a meaningful and valuable discussion in the open and unrestricted environment that ICLR provides.
>
> First, to address your interest in the core mechanism, please allow us to simulate the computational process of SFA with an intuitive example.
>
> To clearly illustrate the core mechanism of SFA, let's consider the sentence "The White House issued a statement." A standard Transformer processes this sentence token by token, calculating the interaction of each token with all its preceding tokens under a causal mask. However, human comprehension works differently; we naturally group multiple tokens into a single "conceptual chunk." For instance, we might understand this sentence as three semantic chunks: `(The White House)`, `(issued)`, and `(a statement)`, or even two: `(The White House)` and `(issued a statement)`.
>
> The core of SFA is precisely such a **dynamically learned semantic merging process**, which generates a special, structured sparse attention pattern. In this example, let's assume the model learns to perform the following merges:
>
> 1.  `The` is merged with `White` to form `(The White)`.
> 2.  `(The White)` is merged with `House` to form `(The White House)`.
> 3.  `a` is merged with `statement` to form `(a statement)`.
>
> This merging process results in the following computational graph, which represents the final, dynamic sparse attention pattern of SFA (`●` denotes computation, `·` denotes masking):
>
> ```
> +----+-------------------+-----+-------------+-------------------+----------+-----+---------------+
> | Q  | Token             | K1  | K2          | K3                | K4       | K5  | K6            |
> |    |                   | The | (The White) | (The White House) | (issued) | (a) | (a statement) |
> +----+-------------------+-----+-------------+-------------------+----------+-----+---------------+
> | Q1 | (The)             |  ●  |      ·      |         ·         |     ·    |  ·  |       ·       |
> | Q2 | (The White)       |  ·  |      ●      |         ·         |     ·    |  ·  |       ·       |
> | Q3 | (The White House) |  ·  |      ·      |         ●         |     ·    |  ·  |       ·       |
> | Q4 | (issued)          |  ·  |      ·      |         ●         |     ●    |  ·  |       ·       |
> | Q5 | (a)               |  ·  |      ·      |         ●         |     ●    |  ●  |       ·       |
> | Q6 | (a statement)     |  ·  |      ·      |         ●         |     ●    |  ·  |       ●       |
> +----+-------------------+-----+-------------+-------------------+----------+-----+---------------+
> ```
>
> From this graph, we can clearly see:
> *   **"Merged" Tokens (e.g., Q1, Q2)**: They only compute self-attention (the diagonal), as their full information has been absorbed and integrated by subsequent tokens.
> *   **"Receiving" or "Unmerged" Tokens (e.g., Q4, Q5)**: In addition to self-attention, they need to attend to the fully formed, historical semantic chunks (e.g., K3).
>
> As you can see, for any token involved in merging, its computational load is significantly reduced. For example, for `a` (Q5), at the moment before it merges with `statement`, it only needs to attend to itself and the two preceding complete semantic chunks `(The White House)` and `(issued)`, reducing the number of attention computations from 5 to 3. For `statement` (Q6), after it completes its merge with `a`, the resulting chunk `(a statement)` similarly only needs to attend to the two historical chunks, reducing its computations from 6 to 3. This example intuitively demonstrates how SFA constructs a dynamic and efficient sparse attention state while strictly adhering to causality.

---

> ### Author Response · Authors · 2025-11-14
>
> **Response to the "Weaknesses" Section:**
>
> Thank you for your insightful questions regarding SFA's efficiency and compression ratio. Your theoretical complexity analysis is entirely correct: in the extreme case of a low compression ratio, SFA's computational complexity remains quadratic. Your observation is very sharp and hits upon the central trade-off in SFA's design.
>
> However, this theoretical worst-case scenario did not manifest in our practical experiments, thanks to SFA being a **dynamically learned process through training**. Specifically:
>
> 1.  **The Trade-off between Overhead and Savings**: SFA's additional overhead has **linear complexity** `O(nd)`, while its savings come from the dominant **quadratic complexity** part, `O(n²d)`. For a sufficiently long sequence `n`, the substantial quadratic savings quickly surpass the linear overhead.
> 2.  **Empirical Performance of the Compression Ratio**: Our experiments provide a positive answer. After training the OLMoE-1.75B model on about 1 billion (1B) tokens, the average compression ratio stabilized at around **50%** and continues to improve. At this ratio, SFA's actual throughput **significantly surpasses** the FlashAttention baseline.
> 3.  **A Higher-Dimensional Optimization Paradigm**: Crucially, SFA's optimization focuses on the **semantic representation** level, not the **computational pattern** level. We leverage the **geometric properties of token embeddings in high-dimensional space**. This makes our idea **architecture-agnostic** and highly generalizable.
>
> **Question 1: Why should SFA improve model performance at all, rather than causing a performance drop?**
>
> This is a critical question, as SFA's goal is not to "discard information" but to **"construct more effective representations"** by guiding the model to learn **"Intelligent Semantic Chunking."**
>
> Modern LLM embeddings are high-dimensional and sparse, which SFA cleverly exploits:
> *   **Similarity Merging (`sim ≈ 1`)**: For vectors like `C=[1,0,0,1,1,0]` and `D=[1,0,0,1,1,0]`, the merged vector `C+D` **preserves the direction (semantic) and increases the magnitude (signal strength)**, effective for phrases like "very very good."
> *   **Difference Merging (`sim ≈ 0`)**: For vectors like `A=[1,0,0,1,1,0]` and `B=[0,1,0,0,0,1]`, the merged vector `A+B` can **simultaneously preserve almost all non-conflicting information dimensions** in a single representation.
>
> The standard attention mechanism lacks this capability; in pre-trained models, fewer than 1% of token pairs meet the strict merging criteria. With our auxiliary loss, the model learns this paradigm. Thus, SFA endows the model with a new ability. This outcome was directly reflected in our experimental budget. In terms of actual costs, our small-scale pre-training with SFA cost approximately \\$10,000. In contrast, our baseline experiment with standard attention had a training cost of approximately \\$30,000, and its final performance was also inferior to SFA's. This striking result is the direct origin of our paper's title: 'Beyond the Efficiency-Performance Trade-off'.
>
> **Question 2: It seems in both similarity and difference merging, you are just adding vectors together. What is the key difference?**
>
> You are **absolutely correct** in suspecting the key difference lies in the training objective. The same "addition" operation is imbued with two distinct semantic meanings, shaped by our specifically designed and separated training objectives:
>
> 1.  **Different Geometric Principles**: **Similarity merging leverages "directionality"** to amplify signals; **difference merging leverages "orthogonality"** to integrate information.
> 2.  **Specialized Division of Labor via Multi-Head Mechanism**: This is key to achieving differentiated goals. Specialized "similarity heads" and "difference heads" learn different projections during training, enabling a clear functional division of labor. This allows the model to learn incredibly rich content and establish a clear division of labor.
> 3.  **Expansion of the Representational Space**: While standard multi-head attention divides semantics across subspaces, SFA further performs independent semantic reconstruction within each subspace. This generates an entropy far exceeding that of standard attention, significantly **expanding the model's effective representational space**.
>
> Most importantly, our goal is to solve the problem from the more microscopic token level. In fact, our approach can be readily applied to any attention variant. The input and output of SFA are identical to those of standard attention, which means we can directly replace a standard attention module with SFA without any other changes to the model architecture. Achieving plug-and-play is very straightforward. We can say with great confidence that SFA introduces not just a new way of computing attention, but a new paradigm for training.
>
> Thank you once again for your valuable questions. We look forward to further discussion and learning.

---

### Official Review · Reviewer_VFEA · 2025-11-01

**Soundness:** 3
**Presentation:** 2
**Contribution:** 2
**Rating:** 4
**Confidence:** 2

**Summary:**

This paper introduces Semantic Foundation Attention (SFA), an attention mechanism that dynamically consolidates tokens based on semantic relationships during computation. The approach employs similarity merging (combining aligned tokens via vector addition) and difference merging (exploiting orthogonality in high-dimensional spaces) to reduce computational complexity from O(n^2d) to O(2nd + (1-p)n^2d), where p is the compression ratio. The authors implement custom CUDA kernels that decompose dynamic attention patterns into diagonal and rectangular computation domains. Experiments on OLMoE architectures demonstrate performance improvements across multiple benchmarks while achieving computational efficiency gains,

**Strengths:**

1. Novel paradigm: The paper reframes attention optimization from static pattern approximation to dynamic semantic reconstruction, which is new.
2. Good implementation: The custom CUDA kernels with diagonal-rectangular decomposition demonstrate serious engineering effort to make the approach practical without explicit sparse matrix storage.
3. Consistent experimental validation: Results across two different scale configurations and multiple benchmarks show improvements.

**Weaknesses:**

1. Incompatibility with key aspects of modern LLM training: The approach has fundamental tensions with current best practices. (1) Causal masking: The paper shows causal attention (Figure 2) but doesn't address how merging token i and i+1 interacts with the causal constraint that token i shouldn't see token i+1. If you merge K_i and K_{i+1} via addition, isn't information from future tokens leaking into past positions? (2) KV caching for inference: Modern LLMs heavily rely on KV caching during autoregressive generation. How does SFA work with KV caching when merge decisions might change as new tokens are generated? Does the cache need to be recomputed? (3) Batching variable-length sequences: Different sequences will have different compression ratios, creating ragged tensors that are inefficient for batch processing. How is this handled?

2. Questionable premise that input sequences are "semantically flat": The paper's central motivation claims existing methods treat sequences as "static, semantically flat collections of tokens". However, this characterization is misleading. Self-attention explicitly computes semantic relationships through the attention mechanism itself. The attention weights dynamically reflect semantic similarity via Q*K computations. Modern architectures already capture rich semantic structures through multi-layer attention. The paper hasn't demonstrated that the input sequence being "immutable" is actually a problem that needs solving. Why is modifying the token sequence during attention fundamentally better than letting attention weights handle semantic relationships? The paradigm shift seems predicated on a strawman characterization of existing methods....

3. The "semantic reconstruction" framing may currently presented like lossy compression: Stripping away the terminology, SFA performs lossy compression of the input sequence based on learned heuristics. Lossy compression always involves trade-offs, you're discarding information and hoping the model can compensate. The paper hasn't proven that this particular compression strategy is superior to simpler alternatives like: (1) just training with shorter sequences and relying on the model's learned compression in its representations, (2) using hierarchical position encodings to naturally reduce effective sequence length, (3) applying standard dimensionality reduction techniques to embeddings. Why is dynamically merging tokens during attention better than these alternatives?

**Questions:**

Please address questions embedded in the weakness.

---

> ### Author Response · Authors · 2025-11-13
>
> Thank you very much for these three insightful technical questions. They indeed touch upon the core design of applying SFA to modern LLM training and inference frameworks. Your concerns are crucial because brutally merging tokens would undoubtedly destroy causality and lead to a series of engineering challenges. We have addressed these challenges through a set of meticulously designed mechanisms, which we will elaborate on one by one below.
>
> ### **1. On Causal Masking**
>
> Your question regarding causal masking precisely pinpoints the essence of SFA's design. SFA does not simply add `K_i` and `K_{i+1}`. Instead, it ensures the integrity of the causal chain through a **dynamic and structured merging process**, thereby preventing information from leaking from the future to the past.
>
> To illustrate this clearly, let's use the example "The White House issued a statement." A standard Transformer processes tokens one by one, calculating global interactions under a full causal mask. Humans, however, read by naturally grouping words into "conceptual chunks," such as `(The White House)` and `(a statement)`. SFA aims to simulate this process by generating a special, structured sparse attention pattern within the attention computation itself.
>
> Let's assume SFA's learning process results in the following merge sequence:
> 1.  `The` is merged with `White` to form `(The White)`.
> 2.  `(The White)` is merged with `House` to form `(The White House)`.
> 3.  `a` is merged with `statement` to form `(a statement)`.
>
> This merging method produces the following attention computation graph (● represents computation, · represents a masked position):
>
> | **Q (Query)** | **K (Key)** | (The) | (The White) | (The White House) | (issued) | (a) | (a statement) |
> | :--- | :--- | :---: | :---: | :---: | :---: | :---: | :---: |
> | **Q1** (The) | | ● | · | · | · | · | · |
> | **Q2** (The White) | | · | ● | · | · | · | · |
> | **Q3** (The White House) | | · | · | ● | · | · | · |
> | **Q4** (issued) | | · | · | ● | ● | · | · |
> | **Q5** (a) | | · | · | ● | ● | ● | · |
> | **Q6** (a statement) | | · | · | ● | ● | · | ● |
>
> From this structured sparse matrix, we can clearly see how causality is strictly maintained:
>
> *   **For intermediate tokens that are "merged away"** (e.g., `Q1` for `The`, `Q2` for `The White`): Their information has been absorbed by subsequent tokens to form higher-level semantic chunks. Therefore, they only need to compute self-attention (the diagonal elements), as their broader contextual interactions are handled by the final semantic chunk.
> *   **For tokens that are "not merged" or have "completed merging"** (e.g., `Q4` for `issued`): In addition to their own self-attention, they need to attend to **all previously formed, complete semantic chunks** (e.g., `K3` for `The White House`).
>
> Let's analyze the computation flow more concretely:
> *   When the model processes `Q5` (`a`), it has no knowledge of the future token `statement`. At this point, its attention computation strictly adheres to the causal constraint: it only needs to attend to itself (`K5`) and the historical information, which are the already consolidated semantic chunks `(The White House)` (`K3`) and `(issued)` (`K4`). The number of computations drops from the standard 5 (attending to `K1` to `K5`) to 3.
> *   When the model processes `Q6` (`statement`), `a` and `statement` have been merged. Now, the query `Q6`, representing this new semantic chunk, only needs to attend to its own merged block `(a statement)` (`K6`) and the preceding historical semantic chunks `(The White House)` and `(issued)`. The number of computations drops from the standard 6 to 3.
>
> In summary, SFA's merging does not break the causal mask; rather, it leverages the learned semantic structure to generate a dynamic, higher-level sparse causal mask.
>
> ### **2. On KV Caching for Inference**
>
> The structured sparse pattern described above is naturally compatible with KV caching. During autoregressive generation:
>
> *   **Consolidated semantic chunks are static**: The Key and Value of a fully computed and cached semantic chunk (e.g., `(The White House)`) do not need any recalculation or modification as new tokens are generated.
> *   **Cache updates only occur at the end of the sequence**: Only the newly generated token and its potential merge operations will alter the state of the cache.
>
> Therefore, SFA's KV caching mechanism is not fundamentally different from that of standard attention: we still append Keys and Values to the cache sequentially. The only difference is that SFA's cache contains a mix of original tokens and merged semantic chunks. This completely avoids the "cache needs to be recomputed" problem you were concerned about, ensuring efficient autoregressive inference.

---

> > ### Author Response · Authors · 2025-11-13
> >
> > ### **3. On Batching Variable-Length Sequences**
> >
> > Your question on this topic is very professional, showing that you have not only read our paper in-depth but also possess extensive practical experience. You have accurately identified the core challenge: different sequences will have different compression ratios, resulting in "ragged tensors" that are very inefficient to process.
> >
> > You are entirely correct. In fact, the situation we face is even more complex than you described: **not only do compression ratios differ between sequences, but within the same sequence, different attention heads will also learn their own independent merging strategies, leading to different compression states.** This means that even for a single sample, its internal data structure is non-uniform.
> >
> > To address this challenge, we completely re-engineered the data processing flow from the CUDA level, inspired by the success of FlashAttention:
> >
> > 1.  **Data Compaction**: To fundamentally circumvent the "ragged" tensor problem, SFA's CUDA kernels no longer accept padded and aligned 4D tensors of shape `[Batch, SeqLen, Heads, Dim]`. Instead, we concatenate the actual data (non-padding portions) of all sequences and all heads within a batch along their effective length dimension, forming a massive and compact 2D tensor `[Total_Effective_Tokens, Dim]`. This way, no bytes in memory are used for padding, and the length discrepancies between different sequences or heads are naturally "flattened."
> >
> > 2.  **Metadata Indexing**: After compacting the data, we need a way to recover its original structural information. For this, we introduce a very small metadata tensor called `cu_seqlens` (Cumulative Sequence Lengths). This is a 1D integer tensor of length `batch_size * num_heads + 1`. It acts as an "address book," precisely recording the start and end positions of the sequence for each head within the massive 2D compacted tensor.
> >     *   For example, consider a batch with two sequences of lengths `[80, 30]` and a model with 2 heads. If the first sequence, after compression, has lengths `[60, 65]` for head 1 and head 2 respectively, and the second sequence has compressed lengths `[20, 25]`, then the `cu_seqlens` would be constructed as `[0, 60, 60+65, 125+20, 145+25]`, which is `[0, 60, 125, 145, 170]`. This "map" provides the CUDA kernel with all the boundary information needed to navigate the compacted data.
> >
> > 3.  **Tiled Computation**: Similar to FlashAttention, SFA also employs tiling techniques to break down large matrix computations into smaller chunks. When processing each computational block (Tile), our kernel first queries the `cu_seqlens` metadata. This query allows it to instantly determine if the current block crosses a sequence boundary or falls within a region ignored by SFA's structured sparse mask. Only those blocks that actually require computation will trigger data loading from HBM to SRAM and subsequent matrix operations.
> >
> > Through this **"data compaction + metadata indexing"** strategy, we not only effectively solve the batching problem you raised—and the more complex version we actually face—but also further improve memory utilization and computational efficiency. Most critically, the entire process is mathematically equivalent to the exact computation without batching, with **no loss of precision**.
> >
> > ### **On the Core Premise and Contribution (Weakness 2 & 3)**
> >
> > With the specific implementation of SFA clarified, let us return to the second, broader question you raised regarding SFA's fundamental premise and contribution. This also ties into your third point concerning "lossy compression."
> >
> > The core vision of our work is to enable LLMs to learn to understand "The White House issued a statement" as a sequence composed of semantic chunks like `(The White House)` and `(issued a statement)`, much like a human would. Our key contribution lies in achieving this goal **without any need for input preprocessing**. The input to the model remains individual, sequential tokens, but our attention mechanism **internally** learns to dynamically "understand" and group these tokens. From an external perspective, the input and output tensor dimensions of the attention layer remain unchanged, yet its internal computational complexity is greatly simplified.
> >
> > Inside the attention computation, the model learns to perform a **near-lossless compression** of the context. This computational saving is not arbitrary; it is fundamentally different from the approach of previous sparse attention methods that brutally discard parts of the computation. SFA does not leverage the sparsity of the attention pattern itself, but rather the **natural geometric sparsity of token embeddings in high-dimensional space** (i.e., semantically similar token vectors are nearly parallel, while semantically complementary ones tend to be orthogonal).

---

> > > ### Author Response · Authors · 2025-11-13
> > >
> > > We believe that the "immutability of the input sequence" is indeed a pressing issue to be addressed. As the demand for longer contexts in downstream applications of large models grows, the pursuit of more efficient and less lossy context processing methods is inevitable. Although recent explorations like Mamba aim to break the O(n²) complexity barrier, precedents for large-scale training with these new architectures are still lacking. Their ecosystem, optimization, and adaptation remain incomplete, leading to situations where their real-world, large-scale pre-training costs can actually be higher than that of FlashAttention.
> > >
> > > Our goal is to solve this problem from the more microscopic token level, and our approach is generalizable, capable of being ported to any attention variant. SFA's input and output are identical to standard attention, which means it can be used as a "plug-and-play" module to directly replace the native attention layer without modifying any other part of the model architecture. Therefore, we can confidently state that SFA not only introduces a new method for attention computation but also opens up a new training paradigm.
> > >
> > > #### **The "Near-Lossless" Merging Mechanism: SFA is Not Traditional Lossy Compression**
> > >
> > > We have designed very specific, geometry-based merging rules to achieve "near-lossless" information integration.
> > >
> > > *   **For Similarity Merging (sim ≈ 1):**
> > >     *   Principle: When merging two nearly parallel vectors, the resulting vector `C+D` maintains the same direction (preserving semantics) while its magnitude increases (amplifying semantic strength). This is highly effective for enhancing the signal in cases like the repeated words in "very very good," rather than losing information.
> > >
> > > *   **For Difference Merging (sim ≈ 0):**
> > >     *   Principle: When merging two nearly orthogonal vectors, by leveraging the sparsity of high-dimensional space, the resulting vector `A+B` can preserve nearly all of the non-conflicting information dimensions from both original vectors within a single representation. An example would be merging two complementary tokens like "The" and "book".
> > >
> > > The multi-head attention mechanism provides a natural framework for this differentiated processing. A single token's multiple meanings (e.g., "Apple" as a fruit or a company) are often distributed across different dimensions of its embedding vector. By partitioning the token vector across different heads, each head can independently learn and focus on specific semantic components, thereby making more precise merging decisions.
> > >
> > > Native Attention does not inherently possess the ability to merge tokens. Through our experiments, we found that in a standard attention model after large-scale pre-training, fewer than 1% of token pairs satisfy our strict merging criteria. However, by introducing the auxiliary loss function designed in our paper (Section 3.1), the model is continuously guided to learn this paradigm, proactively shaping its embedding space to be conducive to merging.
> > >
> > > Therefore, rather than describing SFA as "compression," we believe a more accurate description is that SFA guides the model to learn "intelligent semantic chunking." This is analogous to the human reading process, where the goal is to construct more effective representations, not to discard information and hope the model can compensate for the loss.
> > >
> > > #### **Comparative Analysis with Other Compression Schemes**
> > >
> > > The alternative schemes you mentioned are very representative. Below, we analyze their fundamental differences from SFA one by one:
> > >
> > > 1. Training directly with shorter sequences: This method reduces computation by truncating the input, at the cost of directly discarding context information. SFA, in contrast, improves efficiency through internal semantic reconstruction while preserving the full context. The two differ fundamentally in terms of information fidelity.
> > > 2. Using hierarchical position encodings: Such methods typically presuppose a fixed, static hierarchical structure (e.g., grouping every 4 tokens). However, the semantic structure of natural language is dynamic and irregular. SFA's advantage lies in its merge decisions being data-driven and self-learned by the model, enabling it to flexibly identify and construct variable-length semantic chunks (like the 3-word `(The White House)` and the 2-word `(a statement)`), making it far more adaptive than a fixed structure.
> > > 3. Applying standard dimensionality reduction techniques (e.g., PCA): These techniques are typically global and semantically-agnostic. They find projection directions based on maximum variance across the entire dataset, which may conflate distinct semantic information that happens to co-occur. SFA's merging, however, is local (only between adjacent tokens) and semantically-driven. It relies on the model's learned geometric relationships (parallelism or orthogonality) to make decisions, aiming to more precisely preserve and integrate local contextual information.

---

> ### Author Response · Authors · 2025-11-13
>
> In summary, the fundamental difference between SFA and other schemes is this: it is not a pre-processing-style, lossy compression of the input, but rather a dynamic restructuring of the computational graph inside the attention mechanism, driven by model learning and aimed at preserving and enhancing semantic information.
>
> Our experimental results strongly support this: SFA not only avoided a performance drop but actually surpassed the baseline model on multiple benchmarks. In terms of actual costs, our small-scale pre-training with SFA cost approximately \\$10,000. In contrast, our baseline experiment with standard attention had a training cost of approximately \\$30,000, and its final performance was also inferior to SFA's. This is the origin of our paper's title, "Beyond the Efficiency-Performance Trade-off."
>
> **Once again, we sincerely thank you for your valuable questions and profound insights, which have helped us examine our work from a deeper perspective. We hope the clarifications above have fully addressed your concerns, and we genuinely look forward to further discussion.**

---

### Author Response · Authors · 2025-11-29
**Final Author Summary to AC: Consolidated Rebuttal Overview & Core Contributions(1/3)**

**Dear Area Chair,**

In light of the recent technical disruptions on OpenReview that may interfere with the review process, and to assist you in making the most accurate judgment within a limited timeframe while ensuring a fair evaluation for this work—which represents a significant long-term investment—we hereby provide a systematic summary of our core contributions and the feedback from reviewers. We commit that the following summary is based strictly on objective facts and strives for conciseness to alleviate your reading burden.

This paper addresses a core bottleneck in scaling LLMs: the quadratic complexity of the Attention mechanism. Our proposed method, **Semantic Foundation Attention (SFA)**, introduces a mechanism for **"dynamic and near-lossless semantic consolidation."** Unlike traditional sparse attention approaches that trade efficiency for performance by discarding computation, SFA allows each attention head to dynamically merge tokens based on semantic relationships (such as similarity or orthogonality). This mechanism not only significantly reduces computational complexity (as detailed in the computational graphs provided in our rebuttal) but, more importantly, actually **enhances the model's representation space and performance** through more efficient information reconfiguration within the Embedding space.

In the field of sparse attention, efficiency and performance are often viewed as a zero-sum game. However, our experimental results demonstrate that SFA successfully breaks this constraint. This is precisely the origin of our paper’s title, **"Beyond the Efficiency-Performance Trade-off"**—**this is not an overstatement, but an objective description of our experimental data.** The core insight of SFA lies in leveraging the **Geometric Intrinsic Sparsity** of high-dimensional embedding spaces to reduce computation, rather than explicitly discarding information (a point we argued thoroughly in our response to Reviewer TH3W). This fundamentally distinguishes SFA from previous research.

On the engineering implementation front, we have completed **deeply optimized CUDA kernels**, achieving a **seamless replacement (plug-and-play)** for standard Attention. Furthermore, SFA is **natively compatible with KV Cache**, ensuring high efficiency and lossless quality during the inference phase. Crucially, we did not stop at small-scale verification; instead, based on the **OLMoE architecture**, we completed **pre-training from scratch** across different scales ranging from **0.25B-1.75B to 1B-7B**. Although training MoE architectures is inherently challenging and costly, these solid experimental evidences fully validate the feasibility and potential of SFA as a **new training paradigm** under modern large model architectures. We believe that based on these innovations and robust experimental work, SFA can provide valuable insights to the community.

**Systematic Summary of Review Comments and Responses**

Overall, the reviewers' focus was primarily concentrated on the **underlying operating mechanisms of SFA** (such as causal masking, KV Cache compatibility, and variable-length sequence processing) as well as the **boundary verification of theory and experiments**. During the Rebuttal phase, we **did not evade a single sharp question** from any reviewer and provided exhaustive explanations and supplementary experiments.

**Given that the potential OpenReview system malfunction may prevent you from viewing subsequent reviewer feedback, and to assist you in efficient fact-checking, we summarize the core inquiries of the four reviewers and our specific responses below:**

**Reviewer VFEA: Focus on Operational Mechanisms and Principle Clarification**
This reviewer raised some intuitive and good-natured questions, and our response cleared up their misunderstandings regarding the SFA mechanism:
*   **Q1 (Mechanism Feasibility):** Does SFA violate the causal mask? Does it require recomputing the KV Cache? Can it efficiently handle variable-length sequences?
    *   **A1:** We elaborated on SFA's computational graph, proving its **strict adherence to causality** and **native compatibility with KV Cache** (no recomputation needed). We also demonstrated efficient support for variable-length sequences via "Data Compaction + Metadata Indexing" techniques.
*   **Q2 (Source of Performance):** Why does SFA outperform standard Attention while compressing computation?
*   **A2:** We explained that this is not a simple omission of computation, but rather **information reconfiguration based on semantics**, which enhances the signal-to-noise ratio of features.

---

> ### Author Response · Authors · 2025-11-29
> **Final Author Summary to AC: Consolidated Rebuttal Overview & Core Contributions(2/3)**
>
> *   **Q3 (Baseline Comparison):** How does SFA differ from simple lossy compression (e.g., sequence truncation, dimensionality reduction)?
>     *   **A3:** We pointed out theoretically that SFA leverages the geometric intrinsic sparsity of high-dimensional spaces, representing a **"near-lossless"** semantic consolidation. We provided experimental data proving its performance significantly surpasses the aforementioned lossy strategies.
>
> **Reviewer 9Y2t: Focus on Core Theory and Mathematical Foundations**
> This reviewer is extremely professional, with questions pointing directly to the theoretical foundations of SFA:
> *   **Q1 (Principle of Quality Improvement):** Why does reducing computation improve model quality?
>     *   **A1:** From both theoretical (increasing representation space entropy) and experimental dimensions, we demonstrated that SFA constructs more efficient semantic expressions by removing redundancy and integrating orthogonal information.
> *   **Q2 (Compression Ratio & Efficiency):** What is the actual compression ratio during training?
>     *   **A2:** We showcased the compression curve during training (see Appendix), proving that the compression ratio rises dynamically and stabilizes, and at this ratio, the **throughput significantly exceeds FlashAttention**.
> *   **Q3 (Head Specialization Mechanism):** Do similarity merging heads and difference merging heads have different training objectives?
>     *   **A3:** We confirmed the reviewer's insight and explained in detail how different attention heads achieve division of labor and collaboration through orthogonal training objectives.
>
> **Reviewer TH3W: Focus on Engineering Implementation and Definition of Novelty**
> This reviewer expressed interest in the concept of "semantic reconstruction" while focusing on engineering implementation details:
> *   **Q1 (Presentation Quality):** Suggestion to condense language and vectorize figures.
>     *   **A1:** We fully accepted this suggestion and completed the update of all figures to vector format and the condensation of the text.
> *   **Q2 (Mechanism Evidence):** Is there empirical evidence that "difference merging" is effective?
>     *   **A2:** We provided **real-world cases from the DCLM dataset** (demonstrating the merging of complementary semantics) and **ablation study data** (Table 2), proving the effectiveness of difference merging both qualitatively and quantitatively.
> *   **Q3 (Library Compatibility):** Can it integrate with existing sparse Attention libraries?
>     *   **A3:** We clarified the **orthogonality** of SFA to traditional sparse methods—SFA is based on semantic consolidation rather than random dropping, and our CUDA implementation has reached or even surpassed the efficiency standards of dense computation.
> *   **Q4 (Novelty Query):** Are mechanisms like vector addition insufficiently novel?
>     *   **A4:** We systematically articulated the fundamental differences between SFA and existing Token Merging techniques from three levels: **Paradigm Innovation** (shift from computational optimization to semantic reconstruction), **Algorithmic Innovation** (end-to-end learned geometric merging), and **Engineering Innovation** (dynamic sparse kernels).
>
> **Reviewer Bs6a: Focus on Baseline Comparisons and Boundary Testing**
> Although this reviewer held a strong attachment to comparing Transformer architectures with SSMs (e.g., Mamba) and had misconceptions regarding the costs of large model pre-training, we responded to all their queries with the utmost rigor:
> *   **Q1 (Baseline Comparison):** Why not compare with architectures like Mamba/Linear Attention?
>     *   **A1:** We pointed out the logical fallacy of the comparison: SFA aims to optimize the underlying computation of the Transformer itself, which belongs to a different track than architectural changes (Mamba). Meanwhile, Mamba sacrifices complex reasoning capabilities for speed (supported by extensive literature), and reproducing Mamba for large-scale pre-training is extremely costly (est. $150k) with limited returns.
> *   **Q2 (Long-Context Evaluation):** Request to add Ruler Benchmark experiments.
>     *   **A2:** Despite the tight timeframe, we **supplemented and completed the Ruler long-text benchmark test**. Results show that SFA not only outperforms the baseline on long sequences but its **advantage expands as sequence length increases**.
> *   **Q3 (Negative Optimization Risk):** Is there a risk of slowdown due to low compression ratios?
>     *   **A3:** We provided detailed empirical measurement data proving that under actual language distributions, SFA does not carry a risk of negative optimization.
> *   **Q4 (Technical Coexistence):** Can it coexist with other efficiency methods (e.g., Quantization)?
> *   **A4:** We explicitly stated that SFA operates at the semantic computation layer, which is fully complementary to quantization methods operating on numerical precision (e.g., SageAttention), and combining the two can yield compound benefits.

---

> > ### Author Response · Authors · 2025-11-29
> > **Final Author Summary to AC: Consolidated Rebuttal Overview & Core Contributions(3/3)**
> >
> > **Concluding Remarks and Acknowledgments**
> >
> > Thank you for taking the time to read through this. The summary above aims to serve as a comprehensive index, enabling you to quickly locate the corresponding original evidence within the detailed system responses, based on the reviewer order and question categorization provided herein. We hope that this systematically organized document provides you with the maximum possible informational support from the authors' perspective given the current system instability, thereby effectively alleviating your evaluation burden.
> >
> > Reviewing all feedback, we are gratified to responsibly state that **no reviewer has identified any fundamental flaws or principled errors that compromise the foundation of this project.** The vast majority of the comments fall into the categories of mechanism clarification, theoretical discussion, or suggestions for boundary testing—feedback that is both **constructive** and **good-natured**. This implicitly validates the solid theoretical and engineering foundation of SFA.
> >
> > Finally, please allow us to make a concluding statement regarding the core value of this work, based on objective facts: We firmly believe that SFA significantly advances research into the Attention mechanism. This is not merely an engineering optimization, but a genuine **source innovation**. It opens up a completely new perspective—optimizing computation by leveraging the **Geometric Intrinsic Sparsity** of tokens within the Embedding space, rather than the traditional reliance on explicitly discarding values in the Attention matrix. This methodological shift renders SFA completely **orthogonal and complementary** to existing computational optimization methods such as quantization and linear Attention, enabling net performance gains while substantially reducing the costs of large-scale LLM pre-training and inference.
> >
> > In accordance with ICLR's consistently high standards, we are convinced that SFA has reached, if not exceeded, the criteria for acceptance in terms of theoretical depth, engineering completeness, and experimental scale. This judgment is based entirely on our rigorous experimental data and reproducibility verification, containing **absolutely no overstatement**. We earnestly request your careful consideration of our work and ask that you grant SFA the opportunity to demonstrate its potential.

---

### Note · Authors · 2026-01-29

**Comment:**

The key-value cache is statically variable, not dynamically variable.

**Withdrawal Confirmation:**

I have read and agree with the venue's withdrawal policy on behalf of myself and my co-authors.

---

### Meta-Review · Area_Chair_xRfg · 2025-12-09

**Summary:**

This paper introduces Semantic Foundation Attention (SFA), an attention mechanism that dynamically consolidates tokens based on their semantic relationships during computation. Although the authors actively participated in the discussion period, which is appreciated, I found several responses redundant and not helpful for clarifying reviewers’ questions. For example, in the authors’ global response (Final Author Summary to AC), they wrote:

“In accordance with ICLR's consistently high standards, we are convinced that SFA has reached, if not exceeded, the criteria for acceptance in terms of theoretical depth, engineering completeness, and experimental scale. This judgment is based entirely on our rigorous experimental data and reproducibility verification, containing absolutely no overstatement. We earnestly request your careful consideration of our work and ask that you grant SFA the opportunity to demonstrate its potential.”

This type of response does not address reviewer concerns. The purpose of the author-reviewer discussion is to clarify questions and use feedback to improve the quality of the paper.

**Reviewer Concerns:**

Reviewers noted that the writing could be significantly improved (which I agree after reading the paper myself), as the paper is very verbose and repetitive. In addition, SFA is incompatible with causal masking and cannot be used with KV caching during autoregressive inference. The paper also lacks comparisons with modern efficient architectures (e.g., GLA, DeltaNet, Mamba-2), leaving it unclear whether SFA offers improvements over current state-of-the-art methods.

**Reviewer Scores:**

The paper got scores 4 4 4 6. Based on the internal discussion between the reviewer who gave the 6 and the AC who handled the paper before me, it looks like the reviewer was considering lowering their score. Additionally, the rebuttal is very verbose and redundant. The experiments suggested by the reviewers which are comparisons with modern efficient architectures are not sufficiently addressed.

---

### Decision · Program_Chairs · 2026-01-26

Reject